# AI-Guided Cancer Therapy for Patients with Coexisting Migraines

**DOI:** 10.3390/cancers16213690

**Published:** 2024-10-31

**Authors:** David B. Olawade, Jennifer Teke, Khadijat K. Adeleye, Eghosasere Egbon, Kusal Weerasinghe, Saak V. Ovsepian, Stergios Boussios

**Affiliations:** 1Department of Allied and Public Health, School of Health, Sport and Bioscience, University of East London, London E16 2RD, UK; d.olawade@uel.ac.uk; 2Department of Research and Innovation, Medway NHS Foundation Trust, Gillingham ME7 5NY, Kent, UK; j.teke@nhs.net (J.T.); kusal.weerasinghe@nhs.net (K.W.); 3Department of Public Health, York St John University, London E14 2BA, UK; 4Faculty of Medicine, Health and Social Care, Canterbury Christ Church University, Canterbury CT1 1QU, Kent, UK; 5Elaine Marieb College of Nursing, University of Massachusetts, Amherst, MA 01003, USA; kadeleye@umass.edu; 6Department of Tissue Engineering and Regenerative Medicine, Faculty of Life Science Engineering, FH Technikum, 1200 Vienna, Austria; eghosaseregabriel@gmail.com; 7Faculty of Engineering and Science, University of Greenwich London, Chatham Maritime ME4 4TB, Kent, UK; s.v.ovsepian@greenwich.ac.uk; 8Faculty of Medicine, Tbilisi State University, Tbilisi 0177, Georgia; 9Faculty of Life Sciences & Medicine, School of Cancer & Pharmaceutical Sciences, King’s College London, Strand, London WC2R 2LS, UK; 10Kent Medway Medical School, University of Kent, Canterbury CT2 7LX, Kent, UK; 11Department of Medical Oncology, Medway NHS Foundation Trust, Gillingham ME7 5NY, Kent, UK; 12AELIA Organization, 9th Km Thessaloniki–Thermi, 57001 Thessaloniki, Greece

**Keywords:** artificial intelligence, machine learning, anticancer therapy, patient profiling, migraine, personalized medicine, predictive modeling

## Abstract

Cancer continues to be a leading cause of death globally. Advances in effective treatment have been hindered by difficulties in personalized therapy, especially among patients with comorbid conditions. The use of artificial intelligence (AI) in patient profiling presents a promising strategy for enhancing individualized cancer therapy. AI technologies, such as machine learning (ML), deep learning (DL), and natural language processing (NLP), have become crucial in identifying genetic and molecular biomarkers for cancer and migraine. These technologies facilitate predictive analytics to evaluate the impact of migraine on cancer treatment in patients with comorbidities, helping to forecast outcomes and supporting clinical decision-making through real-time treatment adjustments. AI has considerable potential to enhance the precision and efficacy of managing cancer patients with comorbid migraine. However, challenges related to data integration, clinical validation, and ethical considerations must be addressed.

## 1. Introduction

Cancer remains one of the most significant global health challenges, responsible for a substantial proportion of morbidity and mortality worldwide [1,2]. According to the World Health Organization (WHO), cancer accounted for approximately 10 million deaths in 2020, making it one of the leading causes of death globally [3,4]. Despite the development of advanced therapies, such as targeted treatments, immunotherapies, and precision medicine, the personalization of cancer treatment is hampered by considerable challenges [5,6,7,8]. The complex and heterogeneous nature of cancer, with its diverse genetic, epigenetic, and phenotypic variations, demands individualized and precision treatment strategies [9,10,11]. These challenges are further exacerbated when cancer patients present with comorbid conditions that complicate the management of the cancer and can influence treatment outcomes.

Artificial intelligence (AI) has emerged as a transformative technology in medicine, offering new avenues to enhance cancer diagnosis, therapy, and care by integrating and analyzing complex datasets [12,13]. Applying machine learning (ML), deep learning (DL), and natural language processing (NLP), AI has advanced rapidly and is increasingly being used to tackle challenges in healthcare [14,15,16]. In the context of cancer treatment, AI’s ability to process and analyze large volumes of data, including genomic sequences, medical imaging, and electronic health records (EHR), has the potential to revolutionize patient profiling [12,13]. This process involves identifying the unique characteristics of the primary pathology and comorbid conditions, which is critical for developing the personalized treatment plans that are more likely to be effective and less likely to cause adverse effects.

Patient profiling in oncology involves a comprehensive analysis of various factors that influence cancer behavior, such as genetic mutations, the tumor microenvironment, and patient-specific factors including age, lifestyle, and the presence of other health conditions [17,18,19,20,21]. An in-depth understanding of the specifics of the disease can assist in tailoring more effective and personalized treatment strategies that can significantly improve outcomes [22,23]. This is particularly relevant for patients with comorbid conditions that can affect the trajectory and prognosis of cancer, adding a layer of complexity.

Migraines, which affect a substantial portion of the global population, are a common neurological disorder characterized by recurrent, severe headaches often accompanied by symptoms such as nausea, vomiting, and sensitivity to light and sound [24,25]. The relationship between migraines, cancer and chemotherapy are complex and not yet fully understood. Some studies suggest that migraines may be associated with an increased risk of certain types of cancer, such as breast cancer and glioma [26,27]. Additionally, the presence of migraines in cancer patients can complicate cancer management, particularly in terms of pain management and overall quality of life [28]. Migraines may also impact the efficacy of cancer treatments, with chemotherapy increasing the frequency of headaches, further complicating the treatment landscape [29,30]. Given the high prevalence of migraines and their potential to influence cancer outcomes, there is a growing interest in leveraging AI to improve the profiling of cancer patients who also suffer from migraines [31,32]. In recent years, there has been an increasing use of AI approaches in cancer management and therapy. Figure 1 illustrates a workflow of personalized cancer treatment integrating AI and ML tools. 

AI-based technologies enable the analysis and cross-validation of large sets of multimodal data with a higher level of integration, assisting in generating more precise and individualized data for diagnosis, stratification, and clinical decision-making [33]. Applied to cancer cases with comorbidities such as migraine, this approach could enable better management of the primary pathology and comorbid condition, improving outcomes for patients. Based on the analysis of multi-modal proteomic, metabolomic, genomic and imaging data, the approach facilitates the identification of shared molecular pathways and key mechanisms between migraines and cancer, potentially uncovering new therapeutic targets. It can also help to predict how migraines might influence the progression of cancer or the patient’s response to treatment, allowing clinicians to adjust treatment strategies [34].

This narrative review aims to address several key questions: How can AI be applied to improve the profiling of cancer patients with comorbid migraines? What are the shared genetic, molecular, or clinical markers between cancer and migraines that AI can identify to enhance treatment personalization? How can AI-driven predictive models and decision support systems be integrated into clinical practice to optimize both cancer treatment and migraine management? By exploring these questions, the review seeks to provide a comprehensive understanding of the intersection between AI, cancer, and migraines. It goes beyond a general overview, offering novel insights into how AI can uncover shared mechanisms between cancer and migraines, identify predictive markers, and develop tailored treatment strategies. In doing so, it contributes to the growing body of knowledge by highlighting specific opportunities for leveraging AI in refining anticancer therapy for patients with comorbid migraines. The paper also aims to set future research directions and guide clinical practice by demonstrating how AI can be harnessed to improve outcomes for this complex patient population.

## 2. Methods

### 2.1. Literature Search Strategy

To conduct this narrative review, a comprehensive literature search was performed across multiple databases, including PubMed, Scopus, Web of Science, and Google Scholar. The search was conducted to identify relevant studies published up until August 2024. Keywords used in the search included combinations of the following terms: “artificial intelligence”, “AI”, “machine learning”, “ML”, “deep learning”, “DL”, “cancer therapy”, “oncology”, “patient profiling”, “personalized medicine”, “migraine”, “comorbidities”, and “cancer outcomes”. Boolean operators such as “AND” and “OR” were used to refine the search results. Additionally, reference lists of selected articles were reviewed to identify any further relevant studies not captured in the initial search.

### 2.2. Inclusion and Exclusion Criteria

Studies were included in this review if they met the following criteria: (a) they focused on the application of AI in cancer therapy, particularly in the context of patient profiling or personalized medicine; (b) they addressed the impact of migraines as a comorbidity in cancer patients; (c) they were published in peer-reviewed journals; and (d) they were available in English. Exclusion criteria included, as follows: (a) studies focusing solely on basic science or preclinical research without clinical implications; (b) articles that did not specifically address the intersection of AI, cancer therapy, and migraine comorbidities; and (c) studies with insufficient methodological rigor or unclear outcomes.

### 2.3. Data Extraction, Synthesis, and Quality Assessment

Data from the selected studies were extracted and synthesized to provide a comprehensive overview of the current state of research in the application of AI in anticancer therapy, with a specific focus on migraine comorbidities. Key information extracted included the study design, AI methods used, types of cancer addressed, migraine-related data, and outcomes related to patient profiling and treatment personalization. The extracted data were then analyzed to identify common themes, gaps in the literature, and potential areas for future research. All papers reviewed in this article were scrutinized and selected based on, as follows: (a) the scientific rigor of the presented data with carefully designed experiments and reference to controls; (b) an adequate sample size with a number of replicas and variability; (c) valid statistical analysis and tests with significance of *p* values; and (d) data quality and reproducibility, with discussions limited to peer-reviewed papers only [35].

### 2.4. Narrative Synthesis

The findings from the selected studies were synthesized narratively, with an emphasis on discussing how AI has been applied in the context of cancer therapy and patient profiling, particularly for patients with migraines. The narrative synthesis involved organizing the results into thematic sections such as AI-driven genomic profiling, predictive modeling, and the integration of migraine data into treatment plans. The review also highlighted the novelty and implications of these findings for clinical practice and future research.

## 3. The Role of AI in Anticancer Therapy

AI has become a pivotal technology in the realm of anticancer therapy, transforming how patient care is approached by enabling more precise, data-driven decision-making. Recent advances in AI, particularly in ML and DL, have enhanced the ability of healthcare systems to analyze large volumes and complex datasets rapidly and accurately [14,16]. This capability is crucial in oncology, where the sheer volume of data—ranging from genomic sequences to real-time imaging and patient history—can overwhelm traditional analytical methods [36,37]. AI’s role in anticancer therapy extends beyond basic data processing; it is now integral to the development of personalized treatment plans, predictive modeling of disease progression, and even the identification of novel therapeutic targets.

### 3.1. AI in Personalized Medicine

In the last few years, AI has dramatically advanced personalized medicine, particularly in the context of cancer treatment. By leveraging AI, clinicians can now go beyond standard treatment protocols to develop highly individualized therapy plans tailored to the unique genetic and molecular profiles of cancer patients on a case-by-case basis [8]. Recent developments in AI, such as the use of neural networks to analyze multi-omics data (which include genomics, transcriptomics, proteomics, and metabolomics), allow for the identification of specific biomarkers and genetic mutations that drive cancer in individual patients [38,39]. These insights enable the selection of targeted therapies that are more likely to be effective, reducing the trial-and-error approach that has traditionally characterized cancer treatment.

Moreover, AI’s ability to integrate data from various sources—including EHR, wearable devices, and real-time monitoring tools—allows for the continuous adaptation of treatment plans [40,41]. For example, AI-driven platforms can analyze a patient’s response to a specific therapy in real time, adjusting dosages or switching treatments as necessary to maximize efficacy and minimize side effects. This dynamic approach to treatment is increasingly recognized as a critical factor in improving patient outcomes, particularly in complex cases where standard treatments may fail.

### 3.2. AI in Predictive Modeling

Predictive modeling in cancer therapy has also seen significant advancements with the integration of AI. AI-driven predictive models have become more sophisticated, leveraging ML algorithms that can predict disease progression, potential metastasis, and patient survival with greater accuracy than traditional statistical methods [42]. These models are trained on large datasets, which include not only clinical and genomic data but also lifestyle factors, treatment history, and the presence of comorbid conditions such as migraines.

Recent studies have shown that incorporating data on comorbid conditions into AI models can dramatically improve their predictive accuracy [43]. For instance, a patient with a history of migraines might exhibit different patterns in treatment response or disease progression compared to a patient without such a history. AI can identify these subtle variations, enabling the development of more nuanced treatment strategies that account for the full spectrum of a patient’s health profile [44]. This level of precision is particularly important in oncology, where even small differences in treatment response can significantly impact survival rates.

Furthermore, AI is increasingly being used to predict the likelihood of adverse events or complications during cancer treatment such as the exacerbation of migraines or other neurological conditions. By anticipating these risks, clinicians can preemptively modify treatment plans, such as by adjusting drug regimens or incorporating supportive care measures, to mitigate potential side effects. This proactive approach not only improves patient quality of life but also enhances the overall effectiveness of cancer therapy by ensuring that treatments are better tolerated and thus more likely to be completed.

## 4. Migraine as a Comorbid Condition in Cancer Patients

Migraine, a neurological disorder characterized by recurrent, often severe headaches and associated symptoms, is increasingly recognized as a common comorbid condition in cancer patients [24,45]. The coexistence of migraine and cancer presents unique challenges in clinical management, as the symptoms of migraine can exacerbate the physical and psychological burden of cancer, complicating treatment strategies with potentially detrimental impacts on the outcomes. The dual burden of these conditions demands a comprehensive approach to patient care, where both the oncological and neurological aspects of the patient’s health are addressed. As the understanding of migraine’s impact on cancer patients evolves, it becomes evident that tailored approaches to treatment and patient management are necessary to optimize both cancer therapy and the overall quality of life for these patients. Table 1 provides an overview on how migraines affect cancer patients across various types of cancer, considering demographic factors, the nature of the migraines, potential biological mechanisms, and the impact on cancer treatment.

### 4.1. Epidemiology of Migraine and Cancer

Migraine is one of the most prevalent neurological disorders worldwide, affecting approximately 12–15% of the global population, with a higher prevalence among women [45]. Recent epidemiological studies have explored the relationship between migraine and cancer, revealing potential associations that warrant further investigation [47,49,50]. For instance, some research has suggested that women with a history of migraines may have an increased risk of developing certain types of cancer, such as breast cancer. A study published in recent years has highlighted that the hormonal fluctuations associated with migraines, particularly in women, could be linked to mechanisms that influence cancer risk, although these findings warrant independent verification [51].

Additionally, the role of inflammation, which is a common feature in both migraine pathophysiology and cancer development, has been proposed as a potential link between these two conditions [52]. Chronic inflammation, known to play a role in cancer initiation and progression, may also exacerbate migraine severity, creating a feedback loop that could influence both conditions [53,54]. Understanding the prevalence and impact of migraine among cancer patients is crucial, as this knowledge can inform the development of more comprehensive treatment strategies that consider the interplay between these diseases. While the relationship between migraines and cancer remains complex and not fully understood, the growing body of evidence underscores the need for further research into how they may interact and influence each other.

### 4.2. Impact of Migraine on Cancer Treatment

The presence of migraines in cancer patients can significantly complicate the treatment process and impact the efficacy and tolerability of cancer therapies [55]. Recent studies have shown that migraine sufferers often experience heightened sensitivity to pain and other sensory stimuli, which can exacerbate the side effects of cancer treatments such as chemotherapy, radiation, and immunotherapy [56,57]. For example, the nausea and vomiting commonly associated with migraines may be worsened by chemotherapy, leading to more severe and prolonged episodes that are harder to manage. Additionally, the use of certain chemotherapeutic agents has been reported to trigger or worsen migraine episodes, complicating the overall management of both conditions [55].

Moreover, the cognitive effects of migraines, including difficulty concentrating and memory problems, may interact with the cognitive side effects of cancer treatment, such as “chemo brain”, potentially leading to more pronounced cognitive decline. This interaction can significantly affect a patient’s ability to adhere to treatment protocols and maintain their quality of life during therapy [58]. Recent research has also suggested that migraines might influence the overall prognosis of cancer patients, as the additional neurological burden could affect the patient’s resilience and response to treatment. Given these complexities, there is an increasing recognition of the need for integrated care approaches that address both the oncological and neurological needs of patients. This might include more frequent monitoring of neurological symptoms, the use of tailored pain management strategies that consider the potential for migraine exacerbation, and the incorporation of supportive therapies, such as cognitive behavioral therapy or biofeedback, that have been shown to help manage migraine symptoms. By addressing migraines as a critical component of cancer care, clinicians can better manage the dual burden of these conditions, potentially improving both treatment outcomes and patient quality of life.

## 5. AI in Patient Profiling for Migraine and Cancer

The integration of AI into patient profiling represents a key advancement in the management of complex conditions such as cancer, particularly when complicated by comorbidities like migraines [32,59]. By leveraging AI, clinicians can gain deeper insights into the genetic, molecular, and clinical factors that influence both diseases, enabling more personalized and effective treatment strategies [60]. Figure 2 shows a flowchart diagram summarizing the interaction between migraines and cancer treatment. This approach not only facilitates a better understanding of the shared mechanisms between migraines and cancer but also enhances the precision of predictive analytics and treatment decision-making, ultimately leading to improved patient outcomes. Table 2 provides an overview of how different AI technologies are being applied in the treatment of cancer and migraines, both separately and in combination.

### 5.1. Genetic and Molecular Profiling

AI can revolutionize genetic and molecular fingerprinting by sifting through vast datasets to identify genetic variants and molecular markers that may be common to both cancer and migraines. Recent advancements in AI, particularly in ML and DL, have enabled the identification of shared pathophysiological mechanisms that could underlie both conditions. For instance, certain genetic polymorphisms associated with inflammatory pathways, mitochondrial function, or hormonal regulation may predispose individuals to both cancer and migraines [67,68,69]. AI-driven analyses can uncover these connections by analyzing large-scale genomic and proteomic data, revealing potential therapeutic targets that could be exploited to treat both conditions simultaneously [70].

Moreover, AI can help in the identification of novel biomarkers that predict the onset or progression of migraines in cancer patients, offering opportunities for early intervention [71]. By integrating data from various sources, including genomic sequences, transcriptomics, and even microbiome profiles, AI can provide a comprehensive view of the biological landscape shared between migraines and cancer [71]. This information is invaluable for developing new drugs or repurposing existing therapies to target these shared pathways, potentially leading to more effective treatments with fewer side effects.

### 5.2. Predictive Analytics for Comorbid Conditions

AI-driven predictive analytics offer powerful tools for stratifying cancer patients based on their risk of developing migraines or the potential impact of pre-existing migraines on their cancer treatment outcomes. Through the analysis of historical data, including patient demographics, genetic information, treatment histories, and clinical outcomes, AI can generate predictive models that help clinicians to assess the likelihood of migraine onset or exacerbation during cancer therapy [72]. These models can also predict how migraines might influence cancer prognosis, allowing for more tailored treatment approaches.

For example, if a predictive model suggests that a patient with a specific genetic profile is at high risk for developing severe migraines during chemotherapy, clinicians can preemptively adjust the treatment regimen or incorporate prophylactic migraine therapies to mitigate this risk [24]. Similarly, if the model indicates that a patient’s existing migraines could complicate pain management or lead to increased cognitive side effects, alternative therapies or supportive care strategies can be employed to enhance the patient’s tolerance to the cancer treatment. These predictive capabilities are particularly valuable in creating personalized treatment plans that not only address the cancer itself but also the broader context of the patient’s overall health, including the management of comorbid conditions like migraines.

### 5.3. AI in Treatment Decision Support

AI can be seamlessly integrated into clinical decision support systems (CDSS) to provide real-time, data-driven recommendations for managing patients with both cancer and migraines [73]. These systems are designed to analyze patient data continuously, offering clinicians evidence-based suggestions on how to modify standard treatment protocols to better accommodate the unique needs of each patient. For instance, a CDSS powered by AI can take into consideration a patient’s migraine history, genetic predispositions, and responses to previous treatments, and then recommend adjustments to the current cancer therapy plan.

These adjustments might include changes in medication dosages, the addition of migraine prophylaxis, or the use of alternative therapies that are less likely to trigger or worsen migraines. Furthermore, an AI-driven CDSS can alert clinicians to potential drug interactions or contraindications that might arise when treating cancer patients who are also managing migraines, thereby reducing the risk of adverse effects. Figure 3 illustrates the process of using AI-driven predictive analytics to assess migraine risk in cancer patients and tailor treatment plans, accordingly, aiming to improve treatment outcomes through personalized patient care. By providing these insights in real time, the AI-enhanced CDSS ensures that both cancer and migraine are managed optimally, improving patient outcomes and quality of life.

## 6. Challenges and Future Directions

The integration of AI into patient profiling and anticancer therapy offers significant potential to revolutionize healthcare, particularly for complex cases involving comorbid conditions such as migraines. However, several challenges must be addressed to fully realize AI’s potential in this field. These challenges encompass technical, clinical, and ethical dimensions that require careful consideration and ongoing research.

### 6.1. Data Integration and Privacy Concerns

One of the most significant challenges in applying AI to patient profiling is the integration of multi-modal data while maintaining the strictest standards of patient privacy [74]. The data required for comprehensive profiling—such as genomic information, clinical records, imaging results, and patient-reported outcomes—are often scattered across different platforms and institutions. Integrating these distinct data types into a unified AI model requires sophisticated handling and processing strategies that can seamlessly combine structured and unstructured data from various sources. However, this process raises substantial concerns regarding data privacy and security. As patient data are aggregated and analyzed, the risk of breaches increases, potentially exposing sensitive personal information [75]. Ensuring compliance with data protection regulations, such as the General Data Protection Regulation (GDPR) in Europe and the Health Insurance Portability and Accountability Act (HIPAA) in the United States, is essential [76,77]. Robust encryption methods, de-identification of patient data, and secure data-sharing protocols must be implemented to protect patient confidentiality while allowing for the sophisticated data integration necessary for AI-driven patient profiling [78].

### 6.2. Clinical Implementation and Validation

While AI has demonstrated tremendous promise in research settings, translating these findings into clinical practice remains a significant challenge. AI models developed in controlled environments often struggle to maintain their performance when applied in real-world clinical settings where variability in patient populations, clinical workflows, and data quality can undermine their effectiveness [79,80]. Validating AI algorithms in these diverse and often unpredictable settings is crucial to ensure their robustness, reliability, and generalization [81,82]. This validation process involves extensive testing of AI models across different demographic groups, clinical environments, and disease conditions to confirm that they perform consistently and accurately. Furthermore, clinicians must be trained to interpret and act on AI-driven insights appropriately, which requires a deep understanding of both the AI technology and the clinical context in which it is applied. The development of user-friendly interfaces and clear guidelines for AI use in clinical practice is essential to facilitate its adoption and ensure that AI enhances, rather than complicates, clinical decision-making.

### 6.3. Ethical Considerations

The integration of AI into healthcare also raises significant ethical concerns that must be addressed to ensure its responsible use [83,84]. One of the primary concerns is the potential for bias in AI algorithms. AI systems are only as good as the data they are trained on, and if the training data reflect existing biases—such as those related to race, gender, or socioeconomic status—the AI models may perpetuate or even exacerbate these biases in clinical decision-making [85,86]. Ensuring that AI systems are fair and unbiased requires careful attention to the data used in training, including the inclusion of diverse patient populations and the application of techniques to detect and mitigate bias. Additionally, the transparency of AI-driven decisions is a critical ethical issue. Clinicians and patients must be able to understand and trust the recommendations made by AI systems, which necessitates the development of explainable AI models that provide clear rationales for their decisions. Ensuring that AI systems are not only effective but also fair, transparent, and accountable, is essential for their successful integration into clinical practice.

### 6.4. Future of AI in Management and Therapy of Cancer with Comorbid Migraine

The future of AI in patient profiling for cancer and migraine lies in overcoming the existing challenges while advancing through innovative approaches and interdisciplinary collaboration. To move beyond the current state of AI applications, future research must focus on developing more advanced and explainable AI (XAI) systems that enable clinicians to understand and trust AI-driven recommendations. Unlike traditional “black-box” models, explainable AI provides clear, interpretable outputs, which will be critical for its integration into clinical workflows, particularly for complex cases involving comorbid conditions like migraines. These AI systems should also be capable of handling multimodal data, integrating genomic, clinical, environmental, and even lifestyle factors, while ensuring strict adherence to data privacy through cutting-edge encryption and anonymization technologies.

In addition to explainability, generative AI models represent a promising frontier. Generative AI can simulate hypothetical patient scenarios by predicting treatment responses, side effects, or the progression of both cancer and migraines based on patient profiles. This capability would enable clinicians to experiment with various treatment strategies virtually before applying them in real life, leading to more informed, personalized decision-making. These advancements could significantly improve the precision of treatment plans for patients with cancer and comorbid migraines. Constant validation of AI models in diverse real-world clinical settings will be essential to ensure their robustness and reliability across different populations and healthcare environments. To achieve widespread clinical adoption, AI models must demonstrate consistent performance and generalizability beyond controlled research environments. Future research should prioritize testing AI applications in a variety of demographic and disease contexts to enhance their utility in broader healthcare systems.

Ethical considerations must continue to be one of the primary focus areas as AI evolves in healthcare. To prevent biased outcomes, partiality-detection algorithms and fairness-enhancing techniques must be embedded in AI models from the start. AI-driven systems need to be rigorously audited to ensure that they do not perpetuate healthcare disparities. Additionally, regulatory frameworks must be established to set standards for the transparency, accountability, and auditing of AI technologies in clinical practice. These frameworks should outline specific protocols for evaluating the fairness, safety, and efficacy of AI tools before, and during, their use in patient care.

Moreover, the success of future AI innovations in healthcare will rely heavily on the involvement of all stakeholders, including patients, clinicians, data scientists, and regulators. Their collaboration in the design, implementation, and refinement of AI systems is vital to ensure that these technologies are aligned with the needs and expectations of users. Stakeholder input will ensure that AI systems are user-friendly, effective, and capable of being integrated seamlessly into existing clinical practices.

By addressing these technical, ethical, and regulatory challenges, AI has the potential to revolutionize patient profiling and anticancer therapy. The incorporation of explainable and generative AI models, combined with rigorous clinical validation and stakeholder engagement, will lead to more personalized, effective, and equitable treatment solutions, particularly for complex conditions such as cancer with comorbid migraines. This will mark a significant evolution in the way healthcare is delivered, paving the way for a future where AI-driven personalized medicine becomes a standard of care.

## 7. Conclusions

AI offers transformative potential in anticancer therapy, especially by enhancing patient profiling for individuals with comorbid conditions like migraines. Throughout this review, the integration of AI has been shown to facilitate the analysis of complex datasets, enabling a more tailored approach to treatment that accounts for both cancer and migraine-related factors. AI’s ability to integrate and process genetic, molecular, clinical, and patient-reported data equips clinicians with the tools to create personalized treatment strategies that could significantly improve therapy efficacy, reduce adverse effects, and enhance patient outcomes. By addressing the unique challenges faced by patients with comorbid migraines, AI can enable more comprehensive and nuanced therapy strategies, as discussed in this review.

However, the full potential of AI in this area depends on overcoming several challenges. The integration of diverse datasets, while maintaining privacy, is a critical obstacle. Validation of AI models in real-world clinical environments to ensure their reliability and applicability across diverse patient populations is another area requiring close consideration. Finally, ethical issues, such as the potential for bias and ensuring transparency and fairness, must also be carefully considered. The paper further identifies the importance of addressing these ethical concerns to ensure equitable access to AI-driven healthcare improvements.

Addressing these challenges should enable new opportunities for the broader application of AI in anticancer therapy, particularly for managing complex cases involving comorbid migraines. By doing so, AI can foster more effective, personalized, and equitable cancer treatments, aligning with the goal of improving outcomes for patients with comorbid migraines.

## Figures and Tables

**Figure 1 cancers-16-03690-f001:**
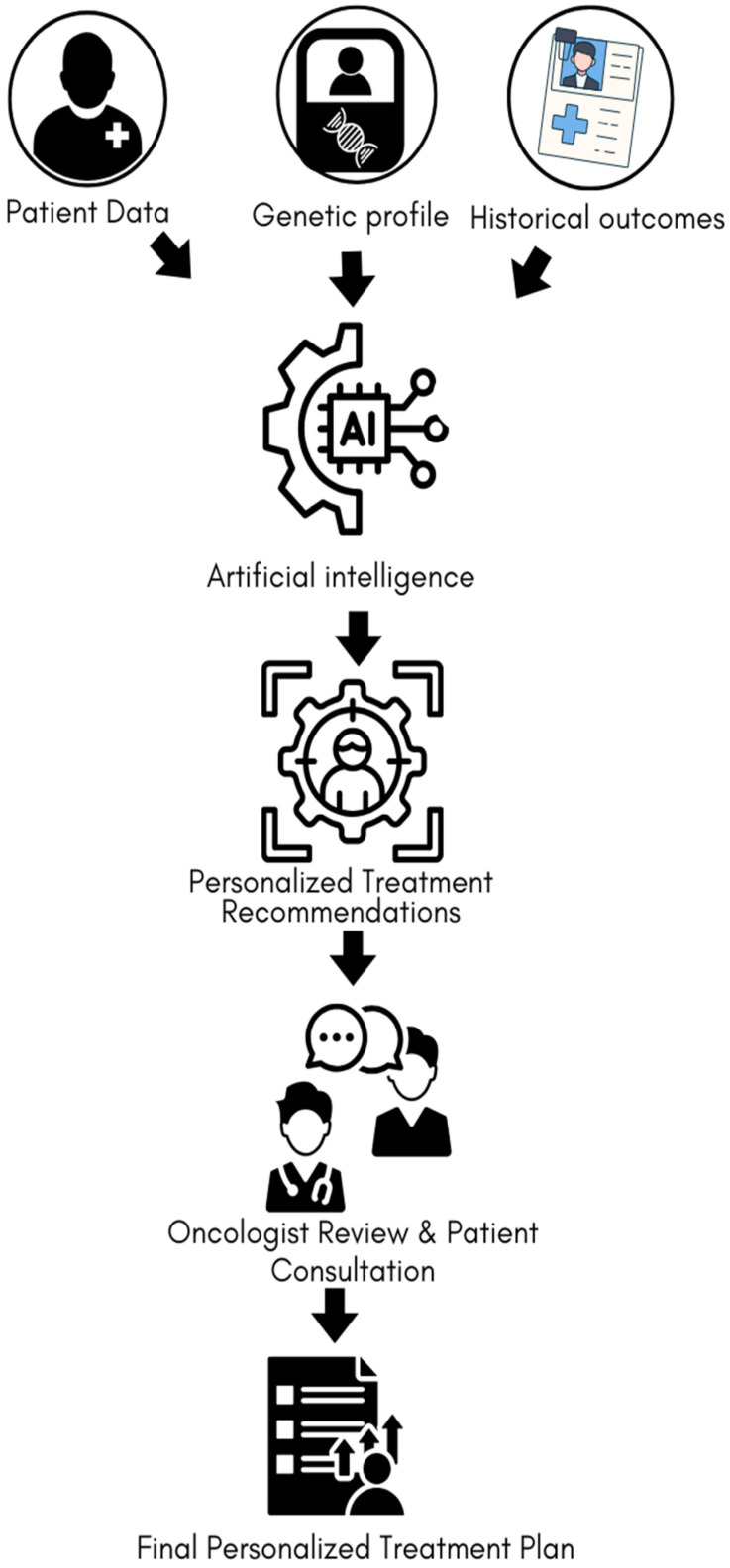
Workflow of personalized cancer treatment using AI and ML. Patient data, genetic profiles, and historical outcomes are processed by AI/ML algorithms to generate treatment recommendations, which are reviewed by an oncologist and discussed with the patient, resulting in a personalized treatment plan.

**Figure 2 cancers-16-03690-f002:**
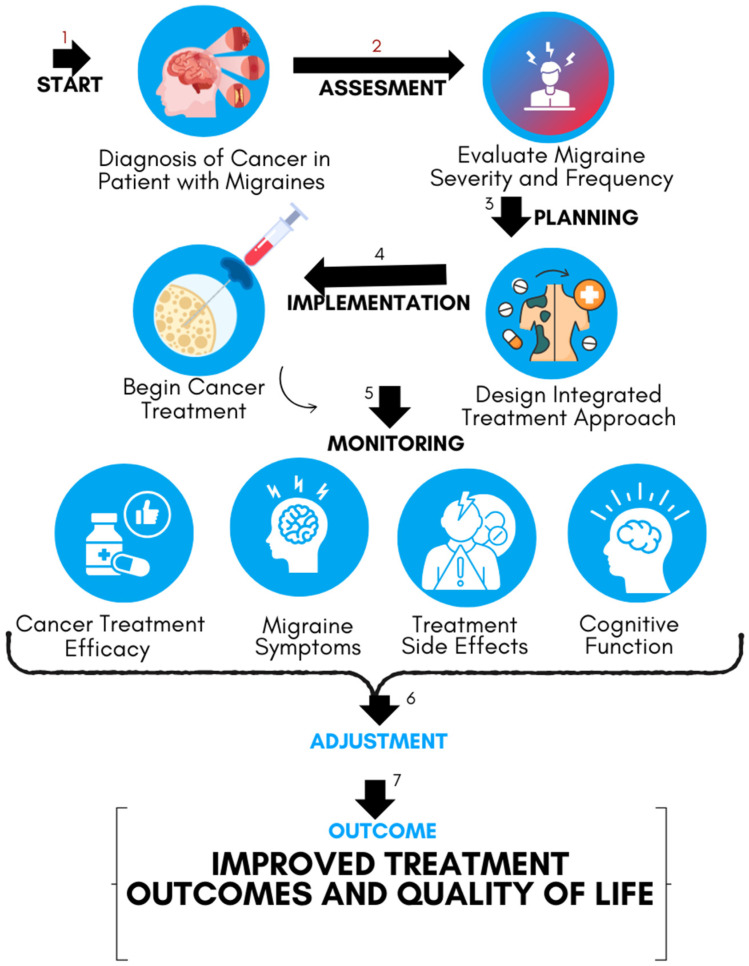
Managing cancer treatment in patients with migraines: from diagnosis through treatment and monitoring, the process aims to improve outcomes and quality of life by addressing cancer and migraine needs together.

**Figure 3 cancers-16-03690-f003:**
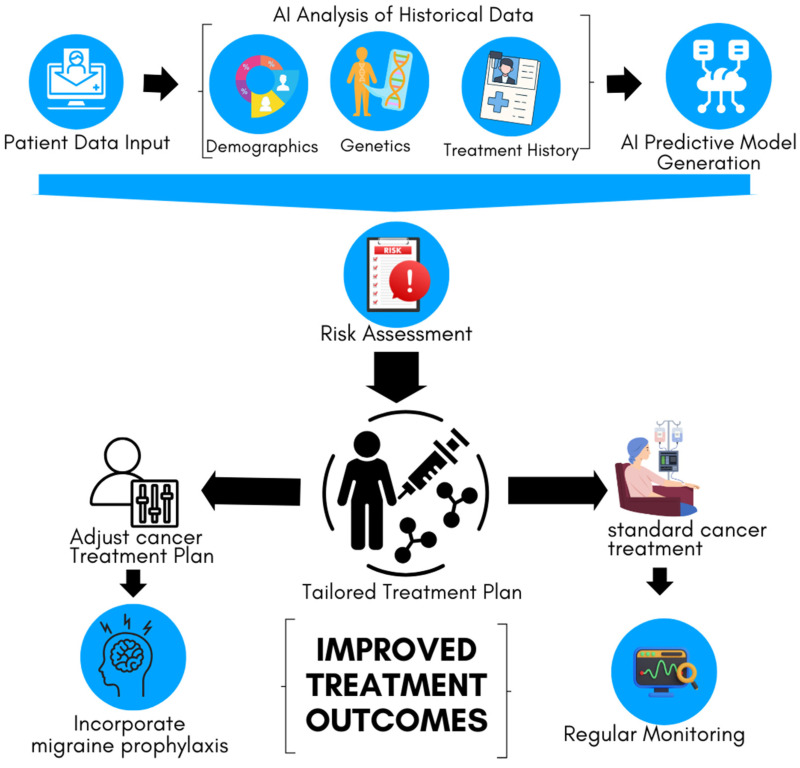
AI-driven personalized cancer treatment plan incorporating migraine management. The AI model generates predictive insights to assess patient-specific risks and guides the development of a tailored treatment plan.

**Table 1 cancers-16-03690-t001:** Prevalence and Characteristics of Migraine in Cancer Patients Across Different Cancer Types.

Cancer Type	Prevalence of Migraine (%)	Study Reference	Population Demographics	Migraine Characteristics	Potential Mechanisms	Impact on Cancer Treatment
Breast Cancer	15–20%	[46]	Female, 54.8 ± 7.2 years	Higher prevalence in ER-positive tumors, often aura	Hormonal fluctuations, estrogen receptor interaction	Increased sensitivity to chemotherapy-induced nausea, the potential need for adjusted hormonal therapies
Glioma	10–15%	[47]	Male-gender, 57.1 ± 16.8 years	Frequent, with aura, often associated with neurological symptoms	Shared genetic markers, inflammatory pathways	Complicates symptom management, especially neurological side effects, possible interaction with anticonvulsants used in treatment
Ovarian Cancer	12–18%	[48]	Female, 25–42 years	Migraine with aura more common, linked to hormonal cycles	Estrogen and progesterone influence, genetic predisposition	Affects response to hormone-based therapies, and increases the need for personalized pain management strategies

**Table 2 cancers-16-03690-t002:** AI technologies used in anticancer therapy and their applications in managing migraine comorbidities.

AI Technology	Application in Cancer Therapy	Application in Migraine Management	Combined Application in Cancer & Migraine	Benefits
ML [61,62]	- Predictive modeling of patient outcomes based on clinical and genomic data - Identification of potential responders to specific therapies	- Prediction of migraine triggers based on patient history and environmental factors - Personalization of migraine management plans	- Predicting interactions between cancer treatments and migraine triggers - Tailoring cancer treatment protocols to reduce migraine exacerbations	- Enhanced treatment precision - Reduction in treatment-related side effects - Improved patient quality of life
DL [63,64]	- Medical image analysis for tumor detection and progression monitoring - Identification of molecular targets for therapy	- Brain imaging analysis to detect migraine-related changes - Pattern recognition in migraine onset linked to neurological activity	- Identifying brain changes predisposing cancer patients to migraines - Monitoring neurological side effects of cancer treatments	- Early detection and intervention for both cancer and migraines - Improved care for neurological symptoms
NLP [63,65]	- Extraction and analysis of unstructured data from EHRs for treatment optimization - Automated literature reviews for identifying cancer therapies	- Parsing migraine symptoms from patient logs - Extraction of migraine triggers and response patterns from unstructured patient data	- Integrating patient-reported outcomes for comprehensive care - Automated tracking of migraine symptoms in cancer patients	- Comprehensive patient profiles - Real-time adjustments to treatment based on feedback
SVM [66]	- Classification of cancer subtypes based on complex biomarker data - Prediction of treatment outcomes based on historical data	- Classification of migraine types and prediction of effective treatments - Analysis of complex datasets to identify migraine triggers	- Classification of patients at high risk for migraine complications - Identifying optimal treatment paths for patients with both diagnoses	- Accurate classification and treatment recommendations - Targeted interventions for high-risk patients
Random Forests [66]	- Ensemble learning for robust prediction of treatment responses and survival outcomes - Stratification of heterogeneous data for patient groups	- Ensemble methods for predicting the most effective migraine treatments - Analysis of patient history and environmental factors to predict migraine risk	- Integrating diverse data sources to predict how migraines may influence cancer treatment - Stratifying patients based on risk factors	- Robust predictions - Better patient stratification - Tailored treatment plans based on comprehensive data integration
Explainable AI (XAI) [65]	- Transparent AI models providing interpretable outputs for treatment decisions - Facilitating clinician trust in AI-driven recommendations	- Explainable models clarifying why certain triggers or treatments work - Improving clinician understanding of migraine management	- Enhancing transparency in combined cancer-migraine treatment protocols - Facilitating the integration of AI in clinical practice	- Better understanding of treatment decisions - Higher clinician trust and adoption - More effective personalized care for complex cases

Abbreviations: AI; artificial intelligence, ML; machine learning, DP; deep learning, NLP; natural language processing, EHR; electronic health records, SVM; support vector machines.

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
