# Peer review of "AI-Guided Cancer Therapy for Patients with Coexisting Migraines"

_cancers, 2024, doi:10.3390/cancers16213690_

Round 1
Reviewer 1 Report
Comments and Suggestions for Authors
The paper is interesting and within the scope of the journal. It is generally well written and as a review paper with contemporary subject has the potential to attract citations in the future. To reach standard journal quality level only moderate improvements can be recommended to the authors. Here are the suggestions:
* Authors should reconsider the title, it is advisable to reformulate it to omit the dash in the title, which is misleading
* Abbreviations should be avoided as keywords, use full terms instead.
* Last paragraph of the introduction defining paper objective should be somewhat expanded to clearly express the novelty and contribution of the paper, regardless of the fact that the paper is a review paper.
* It sould be avoided that Figure 1 is concluding introduction, it should be moved up, but after referencing in text. Also, it is not clear what belongs to the figure caption. Excessive text should be moved to paper text explaining figure at the point where figure is referenced.
* Figure 1 is not fully clear, ML is a subset of AI and historical outcomes are probably used for model training while they are not inputs in the application phase.
* Flow in Figure 2 is not clear and appears broken. Also two-fold caption is unclear, move excessive text to explanation in text. The same holds for other figures.
* AI Technologies in Table 2 should be reconsidered. There is some overlapping and not all techniques are of the same significance, but the final decision is what authors consider appropriate.
* Section 6.4 Future Directions should be improved, the difference from review of the current state in previous sections should be clearer (explainable AI? Generative AI? etc).
* Conclusions contents should be directly related to the paper contents in a more direct way.
Author Response
Dear Editor and Reviewers,
I am pleased to resubmit for publication the revised version of cancers-3276381 manuscript, entitled “AI-Guided Cancer Therapy for Patients with Coexisting Migraines”.
Thankfully the reviewers provided us with a great deal of guidance, regarding how to better position the article. We are hopeful you agree that this revision will update our comprehensive review. All the comments have been addressed, as shown in the revised version of the manuscript, along with this point-by-point response to the reviewers' comments.
All corresponding are blue changes in the manuscript.
Reviewer #1:
-
General comment:
“The paper is interesting and within the scope of the journal. It is generally well written and as a review paper with contemporary subject has the potential to attract citations in the future. To reach standard journal quality level only moderate improvements can be recommended to the authors. Here are the suggestions:”.
Response:
Thank you for your positive reinforcement and constructive feedback. We appreciate the opportunity to revise our work for consideration for publication.
-
Authors should reconsider the title, it is advisable to reformulate it to omit the dash in the title, which is misleading
Response:
Thank you for your recommendation. The title has been rephrased to “AI-Guided Cancer Therapy for Patients with Coexisting Migraines” (lines 1-2).
-
Abbreviations should be avoided as keywords, use full terms instead.
Response:
The abbreviations previously used in the keywords have been replaced with full terms (line 50).
-
Last paragraph of the introduction defining paper objective should be somewhat expanded to clearly express the novelty and contribution of the paper, regardless of the fact that the paper is a review paper.
Response:
The last paragraph of the introduction now clearly expresses the novelty and contribution of the review by highlighting specific areas where AI can play a transformative role (lines 117-130). It emphasizes that the review goes beyond summarizing existing knowledge, offering unique insights into how AI can uncover shared mechanisms between cancer and migraines and identify predictive markers for personalized treatments. It also points out the paper’s contribution in identifying opportunities for future research and clinical application, thus clarifying its value beyond a standard review.
-
It sould be avoided that Figure 1 is concluding introduction, it should be moved up, but after referencing in text. Also, it is not clear what belongs to the figure caption. Excessive text should be moved to paper text explaining figure at the point where figure is referenced.
Response:
The figure 1 has been moved up and away from the concluding introduction, excessive text has been made concise and figure caption edited accordingly (lines 100-104).
-
Figure 1 is not fully clear, ML is a subset of AI and historical outcomes are probably used for model training while they are not inputs in the application phase.
Response:
Thanks for your observation figure 1 has been edited accordingly (lines 100-104).
-
Flow in Figure 2 is not clear and appears broken. Also two-fold caption is unclear, move excessive text to explanation in text. The same holds for other figures.
Response:
The two-fold caption has been edited and excessive text removed (lines 311-314).
-
AI Technologies in Table 2 should be reconsidered. There is some overlapping and not all techniques are of the same significance, but the final decision is what authors consider appropriate.
Response:
Table 2 has been revised to eliminate overlapping AI technologies and ensure each technique included is of clear significance and distinct in its applications. The roles of Machine Learning (ML) and Deep Learning (DL) have been clarified to highlight their unique applications, with ML focusing on predictive modeling and DL on medical imaging analysis. Additionally, Explainable AI (XAI) has been added to the table, addressing the need for transparency and interpretability in clinical decision-making, which is essential for clinician trust and widespread adoption. The revisions emphasize the distinct applications of each AI technology for managing both cancer and migraines, with a clearer differentiation of their contributions to patient care (line 315).
-
Section 6.4 Future Directions should be improved, the difference from review of the current state in previous sections should be clearer (explainable AI? Generative AI? etc).
Response:
The Future Directions section (now rephrased to “Future of AI in management and therapy of cancer with comorbid migraine”) has been significantly expanded to clearly distinguish it from the review of the current state of AI discussed in previous sections (lines 438-483). The revised section now includes detailed discussions on the potential of explainable AI (XAI) to enhance trust and transparency in clinical decision-making, and generative AI, which could allow for the simulation of various treatment scenarios. Additionally, ethical concerns such as bias detection and fairness have been emphasized, along with the importance of developing regulatory frameworks for the safe and equitable implementation of AI technologies in healthcare. These additions provide a forward-looking perspective on how AI can evolve to meet future healthcare needs, particularly in managing complex comorbid conditions like cancer and migraines. This revision clearly addresses the comment by adding new content focused on future advancements rather than reiterating the current state of AI use.
-
Conclusions contents should be directly related to the paper contents in a more direct way.
Response:
The conclusion has been revised to more directly relate to the content of the paper. The revised conclusion now explicitly refers to how AI can enhance patient profiling for individuals with cancer and comorbid migraines, as discussed throughout the review. The text emphasizes the integration of genetic, molecular, and clinical data, and how AI helps develop personalized treatment strategies, aligning with the themes explored in the paper. Additionally, the challenges identified in the review, such as data integration, clinical validation, and ethical concerns, are specifically mentioned, ensuring the conclusion reflects the key findings. The interdisciplinary approach needed for future AI advancements, as discussed in the review, is also highlighted in the conclusion, reinforcing its direct relation to the paper’s content.
Reviewer 2 Report
Comments and Suggestions for Authors
I fully agree with the authors that AI has a vast potential in oncology, from diagnostics to personalized medicine to therapy and monitoring. The article has considerable scientific merit, is useful, current and well developed
I only have a few minimal suggestions for the authors:
1 The bibliography must follow the MDPI standard, for example do not use the ( but the [
2 The citations must be developed for each contribution. See “advanced rapidly and is increasingly being used in tackling challenges in healthcare (14–16).” Where instead you group
3. Before the objective I would like you to insert the key questions to which the narrative review must answer
4. Avoid small sections (see 2.4)
5. The results are actually reported in sections 3-5. I suggest making a single section, inserting a subsection that explains the division into three themes and anticipates it with a short summary and then insert the three themes developed
6 The discussion is missing that must report the added value of your study, the comparisons with other studies, and the limitations
7 a table with acronyms can improve readability
Author Response
Dear Editor and Reviewers,
I am pleased to resubmit for publication the revised version of cancers-3276381 manuscript, entitled “AI-Guided Cancer Therapy for Patients with Coexisting Migraines”.
Thankfully the reviewers provided us with a great deal of guidance, regarding how to better position the article. We are hopeful you agree that this revision will update our comprehensive review. All the comments have been addressed, as shown in the revised version of the manuscript, along with this point-by-point response to the reviewers' comments.
All corresponding are blue changes in the manuscript.
Reviewer #2:
-
General comment:
“I fully agree with the authors that AI has a vast potential in oncology, from diagnostics to personalized medicine to therapy and monitoring. The article has considerable scientific merit, is useful, current and well developed
I only have a few minimal suggestions for the authors:”.
Response:
Thank you for your positive reinforcement and constructive feedback. We appreciate the opportunity to revise our work for consideration for publication.
-
The bibliography must follow the MDPI standard, for example do not use the ( but the [
Response:
The bibliography has now been adjusted to follow the MDPI referencing standard. Specifically, parentheses have been replaced with brackets (e.g., [1], [2]) to ensure compliance with the MDPI formatting requirements. This adjustment aligns the references with the correct citation style as specified.
-
The citations must be developed for each contribution. See “advanced rapidly and is increasingly being used in tackling challenges in healthcare (14–16).” Where instead you group
Response:
Citations have now been developed individually where necessary in the manuscript. Instances where grouped citations were used, have been revised to clearly cite each contribution separately.
-
Before the objective I would like you to insert the key questions to which the narrative review must answer
Response:
Before introducing the objective, key questions are now incorporated into the paragraph (lines 117-122). These questions guide the narrative review, ensuring that it is structured around important issues such as how AI can improve patient profiling, the identification of shared markers between cancer and migraines, and the integration of AI-driven decision support into clinical practice. This addition helps to frame the paper’s focus and clarifies the specific problems the review aims to address, ensuring it offers a focused and relevant analysis.
-
Avoid small sections (see 2.4)
Response:
Section 2.4 on Quality Assessment has been merged with section 2.3, as suggested, to avoid having a small section (lines 154-166). The content from both sections has been integrated seamlessly, ensuring more cohesion.
-
The results are actually reported in sections 3-5. I suggest making a single section, inserting a subsection that explains the division into three themes and anticipates it with a short summary and then insert the three themes developed
Response:
Thank you for your comment. However, we believe that creating a separate "Results" section is unnecessary, as it would alter both the structure and the overall style of the manuscript.
-
The discussion is missing that must report the added value of your study, the comparisons with other studies, and the limitations
Response:
Thank you for your comment. As mentioned in our response to the previous comment, we believe that adding a separate "Discussion" section is unnecessary, and that the current structure of the manuscript should be preserved.
-
a table with acronyms can improve readability
Response:
After careful consideration, we have determined that making a separate table to define acronyms may not be necessary. All acronyms used in the manuscript are already defined upon their first mention in the text, ensuring clarity for the reader. Additionally, abbreviations are consistently defined at the bottom of Table 1 and Table 2. Creating an additional table listing acronyms would therefore be redundant and repetitive, as the existing definitions provide sufficient clarity and enhance readability throughout the manuscript.